# Long-Term Combined Effects of Citrulline and Nitrate-Rich Beetroot Extract Supplementation on Recovery Status in Trained Male Triathletes: A Randomized, Double-Blind, Placebo-Controlled Trial

**DOI:** 10.3390/biology11010075

**Published:** 2022-01-04

**Authors:** José Burgos, Aitor Viribay, Julio Calleja-González, Diego Fernández-Lázaro, Jurgi Olasagasti-Ibargoien, Jesús Seco-Calvo, Juan Mielgo-Ayuso

**Affiliations:** 1Department of Nursing and Physiotherapy, University of León, 24071 León, Spain; joseburgos88@hotmail.com; 2Burgos Nutrition, Physiology, Nutrition and Sport, 26007 Logroño, Spain; 3Glut4Science, Physiology, Nutrition and Sport, 01004 Vitoria-Gasteiz, Spain; aitor@glut4science.com; 4Department of Physical Education and Sport, Faculty of Education and Sport, University of the Basque Country, 01007 Vitoria-Gasteiz, Spain; julio.calleja.gonzalez@gmail.com; 5Department of Cellular Biology, Histology and Pharmacology, Faculty of Health Sciences, University of Valladolid, 42003 Soria, Spain; diego.fernandez.lazaro@uva.es; 6Neurobiology Research Group, Faculty of Medicine, University of Valladolid, 47005 Valladolid, Spain; 7Faculty of Education and Sports, University of Deusto, 20012 Donostia-San Sebastian, Spain; jurgi.olasagasti@deusto.es; 8Health, Physical Activity and Sports Science Laboratory (HealthPASS), Departament of Physical Activity and Sport, Faculty of Education and Sport, University of Deusto, 48007 Bilbao, Spain; 9Physiotherapy Department, Institute of Biomedicine (IBIOMED), University of Leon, Campus de Vegazana, 24071 Leon, Spain; dr.seco.jesus@gmail.com; 10Department of Health Sciences, Faculty of Health Sciences, University of Burgos, 09001 Burgos, Spain

**Keywords:** triathlon, performance, ergogenic aids, muscle fatigue, recovery, hormones

## Abstract

**Simple Summary:**

Recovery is one of the main elements in achieving adequate athletic performance. Various supplements have been used for this purpose. Citrulline (CIT) and Nitrate-Rich Beetroot Extract (BR) are so-called nitric oxide precursor supplements that have shown an ergogenic effect on sports performance when used on a short-term, individual basis. These supplements appear to have other pathways that may promote athletic performance. The purpose of this study was to assess the effect of a co-supplementation for 9 weeks of 3 g/day of CIT plus 2.1 g/day of BR (300 mg/day of nitrates) on recovery by exercise-induced muscle damage markers (EIMD), anabolic/catabolic hormones and distance covered in the Cooper test (CP). Thirty-two male triathletes were randomized into 4 groups of 8 in this double-blind, placebo-controlled trial: placebo group, CIT group, BR group and CIT-BR group. Blood samples and CP were collected at baseline and after 9 weeks. The main conclusions were the combination of 3 g/day of CIT plus 2.1 g/day of BR (300 mg/day of NO_3_^−^) supplementation for 9 weeks did not present any benefit for EIMD. However, CIT-BR improved recovery status by preventing an increase in cortisol and showing an increase in Testosterone/Cortisol ratio and distance covered in the CP.

**Abstract:**

Citrulline (CIT) and nitrate-rich beetroot extract (BR) are widely studied ergogenic aids. Nevertheless, both supplements have been studied in short-term trials and separately. To the best of the authors’ knowledge, the effects of combining CIT and BR supplementation on recovery status observed by distance covered in the Cooper test, exercise-induced muscle damage (EIMD) and anabolic/catabolic hormone status have not been investigated to date. Therefore, the main purpose of this research was to assess the effect of the long-term (9 weeks) mixture of 3 g/day of CIT plus 2.1 g/day of BR (300 mg/day of nitrates (NO_3_^−^)) supplementation on recovery by distance covered in the Cooper test, EIMD markers (urea, creatinine, AST, ALT, GGT, LDH and CK) and anabolic/catabolic hormones (testosterone, cortisol and testosterone/cortisol ratio (T/C)) in male trained triathletes. Thirty-two triathletes were randomized into four different groups of eight triathletes in this double-blind, placebo-controlled trial: placebo group (PLG), CIT group (CITG; 3 g/day of CIT), BR group (BRG; 2.1 g/day of BR (300 mg/day of NO_3_^−^)) and CIT-BR group (CIT-BRG; 3 g/day of CIT plus 2.1 g/day of BR (300 mg/day of NO_3_^−^)). Distance covered in the Cooper test and blood samples were collected from all participants at baseline (T1) and after 9 weeks of supplementation (T2). There were no significant differences in the interaction between group and time in EIMD markers (urea, creatinine, AST, ALT, GGT, LDH and CK) (*p* > 0.05). However, significant differences were observed in the group-by-time interaction in distance covered in the Cooper test (*p* = 0.002; η^2^*p* = 0.418), cortisol (*p* = 0.044; η^2^*p* = 0.247) and T/C (*p* = 0.005; η^2^*p* = 0.359). Concretely, significant differences were observed in distance covered in the Cooper test percentage of change (*p* = 0.002; η^2^*p* = 0.418) between CIT-BRG and PLG and CITG, in cortisol percentage change (*p* = 0.049; η^2^*p* = 0.257) and in T/C percentage change (*p* = 0.018; η^2^*p* = 0.297) between CIT-BRG and PLG. In conclusion, the combination of 3 g/day of CIT plus 2.1 g/day of BR (300 mg/day of NO_3_^−^) supplementation for 9 weeks did not present any benefit for EIMD. However, CIT + BR improved recovery status by preventing an increase in cortisol and showing an increase in distance covered in the Cooper test and T/C.

## 1. Introduction

Prolonged and strenuous exercise produces organic stress [1] that could decrease athletic performance [2,3,4]. As a consequence of this status, there are several alterations in biochemical parameters of exercise-induced muscle damage (EIMD) [5] as well as anabolic/catabolic hormone alterations which could hinder endogenous exercise adaptations [6]. Therefore, in addition to an adequate training program, it could be essential to include different strategies to delay or reduce muscle fatigue and improve adaptation to training [7]. In this sense, supplementation with nitrate-rich beetroot extract (BR) and citrulline (CIT) has been proposed to achieve these goals, partly because they are precursors of nitric oxide (NO) [8,9,10,11].

The NO produces vasodilation by increasing the blood level in muscles and improving their efficiency in muscle contraction and relaxation processes [12]. Moreover, NO regulates force generation and satellite cell activation [13]. In the long term, NO can regulate muscle function and even affect skeletal muscle recovery due to its antioxidant effect and the constant increase in muscle blood flow which, together with an adequate supply of essential amino acids, would allow better muscle fueling [14] and could prevent EIMD [15,16]. Moreover, decreased blood flow to the testis could reduce testosterone synthesis [17]. It has also been shown in animal models that NO enhancement resulted in a significant reduction of ACTH-mediated cortisol production [18]. Consequently, although this mechanism is speculative, increasing NO could improve blood flow in the testis and promote testosterone synthesis by vasodilator effect [14,19] and could be successful in maintaining an anabolic state, decreasing muscular damage and metabolic stress [2,9].

On the one hand, BR supplementation is widely used by athletes as a precursor of NO [20]. When the athletes digest BR, its nitrates (NO_3_^−^) are transformed into nitrites (NO_2_^−^) which are partially reduced to NO by the action of stomach acids and subsequently absorbed in the intestine and passed into the bloodstream [21]. Moreover, BR is rich in other compounds such as phenolic acids, flavonoids, carotenoids and betalains, which have antioxidant effects [22]. Therefore, although the mechanisms for potential improvements in muscle recovery following EIMD after NO_3_^−^ supplementation are not clear, it would be expected that long-term BR supplementation could attenuate EIMD after prolonged, strenuous exercise [22,23] based on the effects of NO and additional compounds. Moreover, long-term BR supplementation could be very beneficial for the maintenance of anabolic/catabolic hormones, as shown by Sarfaraz et al. on testosterone levels [24]. However, short-term BR supplementation (maximum for 3 days) has not presented an improved EIMD and anabolic/catabolic status after a damaging session of eccentric exercise [23] or high-intensity workouts [25], which opens the need for further research.

On the other hand, citrulline (CIT), a non-essential amino acid found primarily in watermelon and produced endogenously by recycling into arginine (ARG) and NO via argininosuccinate synthetase, increases NO availability and its effects [26]. In addition, CIT is an essential element of the urea cycle in the liver [27]. Therefore, it has been suggested that CIT supplementation may eliminate ammonia by urea production [28]. In the same way, CIT is an important activator of muscle protein synthesis in catabolic situations via activation of the mammalian target of rapamycin (mTOR) pathway due to its key role in the regulation of nitrogen homeostasis [29]. Based on these mechanisms, CIT supplementation may favor muscle performance and recovery in different ways, such as activating muscle protein synthesis, improving oxygen distribution to muscle, increasing oxidative ATP production during exercise and phosphocreatine (PCr) during exercise recovery and decreasing blood lactate and ammonium production [14,30,31], which could reduce fatigue and limit EIMD. However, although this proposal would be adequate for athletes, to the best of the authors’ knowledge, there is little research on CIT supplementation in muscle recovery. In this regard, Da Silva et al. [27] did not observe improvements in functional (i.e., number of maximum repetitions, muscle pain and perceived effort), metabolic (i.e., CK and lactate), anabolic (i.e., testosterone and testosterone/cortisol (T/C) ratio) and physiological (electromyographic signal) outcomes of muscle recovery in untrained young adult males after CIT supplementation with 6 g at 60 min prior to the training session. These results of both CIT and BR supplementation on EIMD and anabolic/catabolic hormones may probably be due to the fact that the effects have only been investigated in the short term [28] and under isolated intakes [29,32], suggesting the need to investigate the effects of long-term combination of these two ergogenic aids. In this regard, it has been shown that the effects of some supplements can be synergic when combined over the long term [2,33]. Therefore, it could be considered that the combined effects of CIT (NO precursor and activator of muscle protein synthesis) and BR (NO precursor and antioxidant effect) could reduce EIMD and improve muscle recovery observed by anabolic/catabolic hormone profile [34,35]. This could favor some sporting performance variables [36]. In this sense, the supplementation of 6 g of CIT plus 520 mg of NO_3_^−^ 6 h before the submaximal incremental cycling test has shown improvements in some cardiorespiratory variables, such as VO_2_ [36].

Therefore, the main objective of this research was to assess the effect of the long-term (9 weeks) mixture of 3 g/day of CIT plus 2.1 g/day of BR (300 mg/day of NO_3_^−^) supplementation on recovery status, distance covered in the Cooper test, EIMD markers (urea, creatinine, AST, ALT, GGT, LDH and CK) and anabolic/catabolic hormones (testosterone, cortisol and T/C) in male trained triathletes. The hypothesis was that the combination of CIT plus BR could limit EIMD and improve endogenous recovery observed in lower cortisol and better testosterone and T/C than isolated CIT or BR supplementation.

## 2. Materials and Methods

### 2.1. Participants

Thirty-two male amateur triathletes from the same club (34.37 ± 7.08 years old and 58.79 ± 6.89 mL/min/kg of VO_2max_) with at least 5 years of experience participated in this trial. All athletes rigorously performed the same training methodology, and thus, all of them were exposed to the same training load in terms of type, intensity and duration of exercise (Table 1): 15 h/week, 6 days/week during the 9 weeks. All participants completed a total of 135 h of training during the study.

Likewise, a certified nutritionist (CLR-0020) developed personalized diets for each participant. These diets were planned with the aim of ensuring adequate energy and macro and micronutrient intake considering the training load and the personal features of each triathlete following the international recommendations for an adequate sports performance [37].

All athletes also underwent a medical examination and completed a medical history questionnaire prior to the start of the study to find out whether they had any type of disease and/or injury [38]. The participants did not present any disease, and they did not drink alcohol, smoke or consume other drugs or stimulant substances during the study period which could alter the hormone response. Likewise, to eliminate the probable interference of other nutritional aids with the different outcomes measured in this research, a 2-week washout period was included [39,40,41].

All triathletes were completely informed of all actions of the study and signed a personal statement of informed consent, giving their individual agreement to take part in the proposed work. This trial was considered in accordance with the Declaration of Helsinki (2008) and the Fortaleza update (2013) and was approved by the Human Research Ethics Committee of the University of León, Spain (number: ULE-020-2020). Moreover, this study was registered in clinicaltrials.gov with NCT05143879 number.

### 2.2. Experimental Protocol and Evaluation Plan

This study was planned as a randomized, double-blind, placebo-controlled trial to assess the impact of a 9-week oral supplementation of the combination of CIT plus BR on recovery status by distance-covered performance test, EIMD markers and anabolic/catabolic hormones in this sport population. The proposed doses of CIT (3 g/day) and BR supplements were based on previous scientific studies that found favorable results with similar doses [42,43,44].

The 32 athletes were randomly assigned to four different groups of 8 participants (Table 2) by an independent statistician using the open-source software OxMaR (Oxford Minimization and Randomization, 2014): (I) placebo group (PLG); (II) CIT group (CITG); (III): nitrate-rich beetroot extract group (BRG); and (IV) CIT-BR group (CIT-BRG).

CIT supplementation was included in 3 gelatin capsules of 1 g CIT by Hard Eight Nutrition LLC (7511 Eastgate Rd, Henderson, NV 89011). BR supplementation was included in 3 gelatin capsules of 700 mg (5:1 beetroot extract equivalent to 3500 mg of whole dried root, standardized to contain 0.3% betanin providing 100 mg of NO_3_^−^) by Lindens Health Nutrition (1 Calder Point, Monckton Road, Wakefield, WF2 7AL). The placebo (cellulose) capsules were made of both 1 g and 700 mg being of the same color and shape as the other two supplements to avoid the placebo effect [45]. All athletes took the same number of capsules per day (3 capsules of 1 g (BIG) and 3 capsules of 700 mg (SMALL)) based on their groups: PLG: 3 BIG of cellulose + 3 SMALL of cellulose; CIT: 3 BIG CIT and 3 SMALL of cellulose; BR: 3 BIG of cellulose + 3 SMALL BR; and CIT-BRG: 3 BIG CIT + 3 SMALL BR. In order to ensure blinding, all BIG capsules were white (CIT and placebo) and all SMALL capsules were red (BR and placebo).

All participants took 3 BIG and 3 SMALL capsules, either the placebo or aids, during the 7 days of the week after each of the 3 main meals (1-1-1) to eliminate any influence of circadian variation [46]. Athletes were informed that they should not brush their teeth or rinse their mouths for 2 h after the intake of the capsules, based on the effect of oral bacteria on the reduction of NO_2_^−^ from NO_3_^−^. In addition, they were unaware of the contents of the capsules provided to them weekly by an independent nutritionist (LR003) who confirmed that all triathletes complied with the intake protocol.

### 2.3. Blood Collection

All triathletes arrived at the laboratory at 8:30 a.m. for blood extraction at two different moments during the trial: (T1) at baseline and (T2) after 9 weeks of supplementation. For the evaluation/assessment of EIMD and hormonal outcomes at T1 and T2, antecubital venous blood samples were collected. All samples were obtained after at least 12 h of fasting and 48 h without any previous exercise and after being at rest for 30 min.

The EIMD markers (urea, creatinine, AST, ALT, GGT, CK and LDH) were measured using the Hitachi 917^®^ automatic autoanalyzer (Hitachi Ltd., Tokyo, Japan) [47]. Serum hormone outcomes (total testosterone and cortisol) were measured using an enzyme-linked fluorescent assay with the aid of a multiparametric analyzer (MINI VIDAS^®^, Biomerieux, Marcy l’Etoile, France) [3]. The substrate 4-methylumbelliferone was used, and fluorescence emission was performed at 450 nm and, after stimulation, at 370 nm [48]. The intra-assay CV was 5.7%, and the CV of the intermediate assay was 6.2%. Finally, T/C was considered by dividing testosterone by cortisol.

### 2.4. Cooper Test

After blood analysis and 2 h after the standardized breakfast (2 g of CHO/kg BM and consisting of rice, corn cereal with oat beverage, cooked fruit and biscuits with jam or sweet quince, cheese or paste) [37], the athletes performed a Cooper test. Athletes were familiar with this test given that they usually use this test throughout the season.

Before starting the test, a standardized 15 min warm-up was performed: 8 min incremental run; 3 min of core work; 2 min of trunk, hip and leg muscle exercises; and 2 min of different types of jumps. The Cooper 12 min run test was conducted under the observation of the research team on a 400 m synthetic sports track. The participants completed the traditional test protocol, which consisted of covering the farthest feasible distance in 12 min [49]. The total distance covered in this time was measured immediately after the test was completed using markers placed on the track at 50 m intervals [50].

### 2.5. Anthropometry

The same internationally certified anthropometrist (ISAK level 3 with certificate number: #636739292503670742) performed the anthropometric measurements for all triathletes based on the International Society for the Advancement of Kinanthropometry (ISAK) protocol [51]. Height (cm) was obtained by a SECA^®^ measuring rod (Mod. 220; SECA Medical, Bradford, MA, USA), with 1 mm precision. Body mass (kg) was measured using a SECA^®^ model scale (Mod. 220; SECA Medical, Bradford, MA, USA), with 0.1 kg precision. Body mass index (BMI) was considered by the equation body mass/height^2^ (kg/m^2^). Six skinfolds (mm) were assessed—triceps, subscapular, supraspinal, abdominal, front thigh and medial calf by a Harpenden^®^ Skinfold Caliper (Harpenden Skinfold Caliber, British Indicators Ltd., London, UK) with 0.2 mm precision—and the sum of all of them was considered. Girths (cm) (relaxed arm, flexed arm, minimum waist, 1 cm below the buttock thigh, mid-thigh and calf girth) were measured with an inextensible metallic Lufkin^®^ measuring tape model W606PM (Cooper Tools, Apex, NC, USA) with 1 mm precision. Fat mass (FM) and muscle mass (MM) were estimated by the Carter and Lee equations, respectively [52].

### 2.6. Dietary Assessment

The nutritionists participating in the study (J.B.-B. and J.M.-A.) informed all triathletes about proper food tracking. They tutored the participants regarding 2 validated methods of dietary recall [51]. The first method was a food frequency questionnaire (FFQ) previously used in other sport populations [53] which triathletes should complete at T2. The athletes should recall their average food “frequency” intake based on certain food groups over the previous 9 weeks. Food frequency was based on the number of times each food was consumed per day, week or month. The serving sizes consumed were estimated through the standard weight of food items or by determining the portion sizes by looking at a book containing over 500 photographs of food [54]. Energy (kcal) and macronutrient (g) consumption was determined by dividing the reported intake by the frequency in days using a validated software package (Easy diet^©^, online version 2020) [55]. The total energy and macronutrient intake per kilogram of body mass was calculated for each athlete. The second method was a seven-day dietary recall collected the week prior to T1 and during the week of T2. This method was used to check if the results of the FFQ were similar to those of this recall [52].

### 2.7. Statistical Analysis

The data are shown as means and standard deviations. The Shapiro–Wilk test (*n* < 50) was used to determine normality. Likewise, the homoscedasticity assumption was tested with the previous Levene test. Thereafter, differences from T1 to T2 in each group separately were assessed using Student’s *t*-tests for parametric paired data. Then, a two-way repeated-measures analysis of variance (ANOVA) test was performed to assess the interaction effects (time × supplementation group).

On the other hand, the percentage changes of the outcomes studied between T1 and T2 in each study group were calculated as Δ (%): ((T2 − T1)/T1) × 100). A one-way ANOVA test was performed to determine if there were significant differences between the means of the different outcomes analyzed among the 4 study groups. A Bonferroni post hoc test was applied for pairwise comparisons among supplemented groups to establish statistical significance levels.

Effect sizes as a qualitative measure were estimated by partial eta squared (η^2^*p*). Given that this measure overestimates effect sizes, the values were interpreted based on Ferguson, who indicated no effect if 0 ≤ η^2^*p* < 0.05, minimum effect if 0.05 ≤ η^2^*p* < 0.26, moderate effect if 0.26 ≤ η^2^*p* < 0.64 and strong effect if η^2^*p* ≥ 0.64 [56].

The analyses were completed by SPSS^®^ software version 24.0 (SPSS, Inc., Chicago, IL, USA) and Microsoft Excel^®^ version 24 (Microsoft Corporation, Redmond, WA, USA) and graphics using GraphPad Prism 6 software (GraphPad Software, Inc., San Diego, CA, USA). Statistical significance was designated when *p* < 0.05.

## 3. Results

During the trial, the triathletes did not present significant statistical differences (*p* > 0.05) in energy and macronutrient intake values among groups (Table 3).

Body mass, BMI, muscle mass and fat mass percentage did not present significant differences (*p* > 0.05) in the interaction group-by-time (Table 4).

Figure 1 shows the distance covered in the Cooper test at both T1 and T2. Significant differences can be seen in the group-by-time interaction in this parameter (*p* = 0.002; ƞ^2^*p* = 0.418). In addition, significant increases (*p* < 0.05) were observed between study moments in distance covered (T1: 2953.1 ± 372.7 vs. T2: 3079.6 ± 423.5 m) in CIT-BRG.

The EIMD markers did not present significant differences (*p* > 0.05) in the group-by-time interaction (Table 5). However, significant differences were observed between T1 and T2 for BR in creatinine (T1: 0.92 ± 0.11 vs. T2: 0.88 ± 0.09 mg/dL; ƞ^2^*p*: 0.063) and LDH (T1: 445.38 ± 247.59 vs. T2: 393.88 ± 63.37 UI/L; ƞ^2^*p*: 0.083).

Table 5 displays significant differences in the group-by-time interaction for cortisol (*p* = 0.044; ƞ^2^*p* = 0.247) and T/C (*p* = 0.005; ƞ^2^*p* = 0.359). In this sense, a significant difference was observed for T/C in CIT-BRG with respect to PLG at T2. On the other hand, a significant decrease in testosterone levels and T/C was observed in PLG, CITG and BRG after 9 weeks of supplementation (Table 6).

Figure 2 shows the percentage change in distance covered in the Cooper test for each of the study groups. Significant differences can be observed in this parameter (*p* = 0.002; ƞ^2^*p* = 0.424). Concretely, CIT-BRG presented a significantly higher value in the % change than PLG and CITG (*p* < 0.05).

Figure 3 indicates significant differences in cortisol percentage change (*p* = 0.049; η^2^*p* = 0.257) between PLG and CIT-BRG. Moreover, T/C percentage change presented statistically significant differences (*p* = 0.018; η^2^*p* = 0.297) between CIT-BRG and PLG. In the case of testosterone, there were no significant differences among groups in percentage change (*p* = 0.149).

## 4. Discussion

This study was planned to assess the effect of long-term (9 weeks) combination of 3 g/day of CIT plus 2.1 g/day of BR (300 mg/day of NO_3_^−^) supplementation on recovery status by distance covered in the Cooper test, serum EIMD markers and testosterone and cortisol in male triathletes. The EIMD markers (urea, creatinine, AST, ALT, GGT, LDH, CK) did not show any significant differences in the group-by-time interaction. However, triathletes showed a significantly better group-by-time interaction in distance covered in the Cooper test and anabolic/catabolic hormone status in CIT-BRG by preventing an increase in cortisol and a better T/C ratio. Furthermore, while CITG and BRG showed a significant decrease in testosterone levels, CIT + BR supplementation prevented a decline of this anabolic hormone. These significant results could be motivated by the synergistic effect that both supplements provided on the variables used to determine recovery status.

The balance between training loads and recovery are key factors in improving athletic performance [4]. To assess and control this balance, and with the intention of avoiding fatigue and maintaining an adequate performance, there are numerous parameters utilized, such as EIMD markers and anabolic/catabolic hormones [57,58]. Although there is an acute intensification of EIMD markers after exercise [2,59], long-term maintenance of high EIMD values could indicate a chronic fatigue status and inadequate adaptation to training [60]. In addition, it has been observed that anabolic/catabolic hormone status is changed after exercise due to an acute effect [58,61]. However, long-term variations in these hormones may be indicators of an adequate endogenous adaptation or, on the contrary, a fatigue status and, therefore, of an impaired sports performance [6]. Testosterone is an anabolic and androgenic hormone secreted by the hypothalamic–pituitary–testicular axis, and its increase specifies an overall anabolic state [62]. Nevertheless, cortisol, secreted by the hypothalamic–pituitary–adrenal axis, is a steroid hormone considered as a factor that indicates accumulated stress, and therefore, its increase suggests an accumulation of stress or catabolism [63]. Consequently, an increase in testosterone and/or a decrease in cortisol would lead to an increase in the testosterone/cortisol ratio, as an indicator of adaptation to training, thus indicating better endogenous recovery, while a decrease would indicate fatigue status [61,64]. In order to achieve these effects, some supplements that promote the NO pathway, such as CIT and BR, have been proposed [31].

It has been shown that NO can enhance recovery status through certain mechanisms [65], such as increasing protein synthesis through vasodilation of the arteries and veins of skeletal muscle that improve nutrient flow to the muscles, which in the long term favors muscle growth and repair [66]. In addition, it has been suggested that NO probably promotes angiogenesis in tissues by regulating the expression of the vascular endothelial growth factor [67]. Moreover, it has been demonstrated that skeletal muscle has the capacity to store, transport and metabolize NO_3_^−^ and NO_2_^−^ [68]. Therefore, chronic supplementation with NO precursor supplements (CIT and BR) would increase the levels of NO_3_^−^ stored in skeletal muscle that is beneficial for NO production [69]. All these mentioned mechanisms could probably work in a complementary manner by enhancing endogenous recovery. A more efficient production of energy during exercise would reduce fatigue and thus decrease EIMD through an increase in protein synthesis [70]. This improved regeneration would lead to a decrease in stress and thus a reduced catabolic state, which would be reflected in anabolic/catabolic hormones [11].

In addition to the effect on NO, the CIT has been found to stimulate muscle protein synthesis by activating mTOR through the PI3K/MAPK/4E-BP1 pathway [71] and by increasing ARG production, which promotes growth hormone secretion [72]. Likewise, increased ARG production will improve intramuscular creatine levels, which will also allow an increase in phosphocreatine reserves, contributing to energy supply through a more efficient ATP regeneration and lowering fatigue, resulting in a decrease in EIMD after a high-demanding training [31]. Moreover, being part of the urea cycle, CIT facilitates the functioning of this cycle, helping to reduce the accumulation of ammonium and blood lactate concentration, improving the clearance capacity of these substances and, therefore, reducing the fatigue caused by their accumulation [73].

To the authors’ knowledge, the effects of long-term combination of CIT plus BR supplementation on EIMD markers have not been studied in depth [74]. In this sense, the present trial did not present significant differences in the interaction between group and time in EIMD markers (urea, creatinine, AST, ALT, GGT, LDH and CK) (*p* > 0.05; Table 4). In the same line, some investigations that have evaluated the acute effects of these supplements individually have not found improvements in EIMD markers. Daab et al. did not find significant differences in CK and LDH before the Loughborough Intermittent Shuttle Test between the supplemented and the placebo groups after 7 days (3 days pre-exercise, test day and 3 days post-exercise) with 150 mL of BR juice (250 mg of NO_3_^−^) taken in two intakes per day (08:00 and 18:00 h) in soccer players [74]. Likewise, Martínez-Sanchez et al. did not present significant differences in the biochemical markers AST, ALT and CK between the supplemented group and placebo with CIT-enriched watermelon juice (3.45 g per 500 mL/day) taken two hours before a half-marathon race [66]. Therefore, considering that the results of this study did not offer any beneficial effects on the EIMD markers, the results obtained by the combination of CIT plus BR in the present study dismantle the original hypothesis in which it was predicted that both supplements could work in a complementary manner by reducing EIMD.

Although, to the authors’ knowledge, the effects of long-term combination of CIT plus BR supplementation on anabolic/catabolic hormones have not been studied, in the current study, the combination of these supplements showed a better group-by-time interaction in distance covered in the Cooper test and anabolic/catabolic hormone status in CIT-BRG (Table 5 and Figure 3) by preventing an increase in cortisol (*p* = 0.044; η^2^*p* = 0.247) and a better T/C ratio (*p* = 0.005; η^2^*p* = 0.359). Furthermore, while CITG and BRG showed a significant decrease in the testosterone level after 9 weeks (*p* < 0.05), CIT + BR supplementation prevented a decline in this anabolic hormone. Nevertheless, some authors have shown the effect of both supplements individually on these hormones. In this way, Da Silva et al. did not observe improvements in the T/C ratio during the recovery period at 24, 48 and 72 h post-exercise in untrained young adult men after 6 g CIT supplementation before a 60 min workout [27]. These authors indicated that the inability to improve anabolic factors results in no beneficial effect of CIT supplementation on muscle regeneration during an acute recovery period. However, the chronic changes in cortisol and testosterone can be related to accumulated stress and body regeneration during the sports season [6]. In this way, Garnacho-Castaño et al. showed that BR supplementation did not appear to influence anabolic/catabolic status in response to acute high-intensity workouts after drinking 140 mL of BJ (~12.8 mmol NO_3_^−^) [25]. On the contrary, in this study, CITG and BRG presented a maintenance of distance covered in the Cooper test and a decrease in testosterone levels and T/C after 9 weeks of supplementation. These data could indicate an inadequate recovery status in these groups because after 9 weeks with adequate training, a better sports performance would be expected. Nevertheless, this study presented a significantly better recovery status in CIT-BRG represented as an increase in distance covered in the Cooper test and maintenance of testosterone and T/C ratio after 9 weeks of combined supplementation. These adaptations were obtained by preventing an increase in cortisol and/or a decline in testosterone in CIT-BRG with respect to other supplementation groups.

In this sense, CIT is a key activator of muscle protein synthesis in catabolic situations, such as high-intensity training periods, via activation of the mTOR pathway due to its key role in the regulation of nitrogen homeostasis [75]. Likewise, testosterone increases mTOR pathway [76] and cortisol inhibits mTOR pathway signaling [75]. Thus, increasing testosterone and controlling cortisol secretion could result in lower stress and adequate muscle regeneration [6]. In this sense, the long-term effect of CIT enhancing NO could increase blood flow in the testis promoting testosterone synthesis [19] and maintaining the testosterone level by vasodilator effect [77]. In addition, the enhancement of NO reduces ACTH-mediated cortisol production [18]. Consequently, although this hypothesis is speculative, increasing NO could be successful in maintaining an anabolic state, decreasing metabolic stress [2,9]. Therefore, the results obtained in the CIT-BRG group could show how independent pathways in muscle recovery (NO and mTOR) can be synergistically activated with both supplements to obtain better results.

### 4.1. Limitations, Strengths and Future Research

It should be noted that it is difficult to obtain larger samples in athletes as not many of them have the availability to comply with the training and supplementation instructions required by the study. In addition, the effects that both supplements used could have on the muscle were speculative because no evaluation was included in this regard. On the other hand, sampling using a convenient, non-probabilistic sampling procedure may produce results that are not representative of the rest of the population. These limitations may underrepresent the results and may affect study outcomes. For this reason, the results should be considered in the context of the study. However, the methodology used in this trial, a double-blind, placebo-controlled trial, is the most important strength. In addition, another strength was the control of the triathletes’ diet, as well as the control of the body composition throughout the intervention process, so that these outcomes did not influence the final results. Another strength is the synergistic potential of the study.

Future research should continue to study the long-term effects of this combination on recovery, using different markers, such as sports performance, in order to expand the existing knowledge on this combination. It should also examine the effectiveness of these supplements in athletes who have already been diagnosed with an overtraining state to determine whether the use of these supplements as part of treatment would accelerate recovery. In addition, it should analyze how this potential combination affects the female population or anaerobic sports, given that this study only focused on males and measured aerobic performance.

### 4.2. Practical Application

This research could be of interest to physicians and nutritionists who want to achieve better post-exercise recovery for their athletes. Considering that 3 g/day of CIT plus 2.1 g/day of BR (300 mg/day of NO_3_^−^) for 9 weeks could advance muscle and endogenous recovery, supplementation phases could be considered in the intensive training phases.

## 5. Conclusions

In conclusion, although the combination of 3 g/day of CIT plus 2.1 g/day of BR (300 mg/day of NO_3_^−^) supplementation for 9 weeks did not present any benefit for EIMD, it prevented an increase in cortisol and a decline in T/C compared with placebo or isolated supplementation. Moreover, this combination promoted a better distance covered in the Cooper test after 9 weeks of supplementation. Therefore, the combined use of 3 g/day of CIT and BR (300 mg/day of NO_3_^−^) could promote a faster muscle recovery status but without preventing EIMD.

## Figures and Tables

**Figure 1 biology-11-00075-f001:**
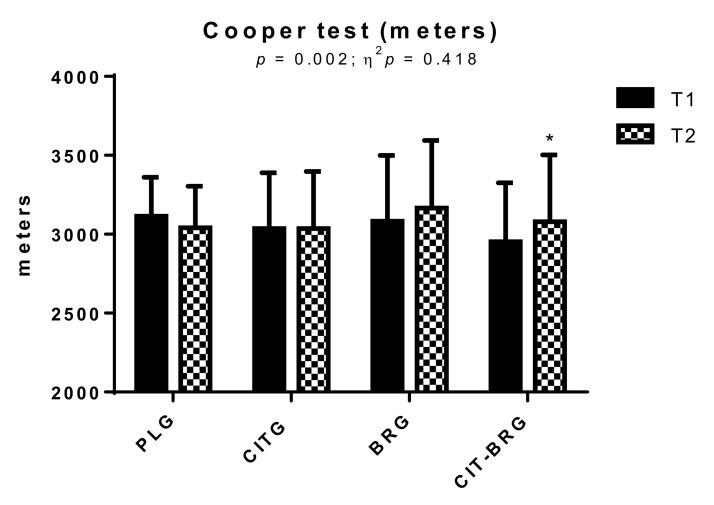
Distance covered in Cooper test by triathletes at T1 and T2 (after 9 weeks of supplementation). Data are presented as mean ± standard deviation. *: Significant differences with respect to T1. *p* < 0.05.

**Figure 2 biology-11-00075-f002:**
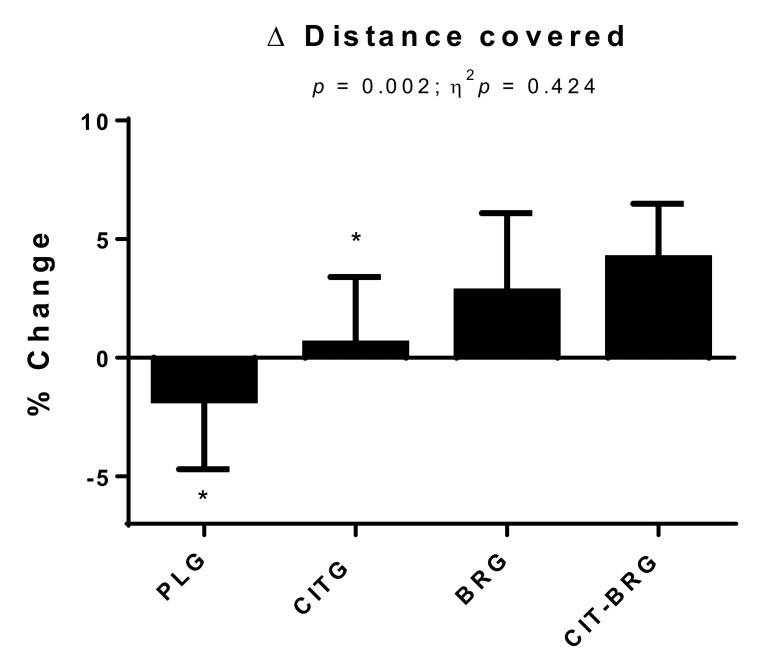
Percentage changes during the study in estimated distance covered in Cooper test in groups. Data are presented as mean ± standard deviation. Δ: ((T2 – T1)/T1) × 100. *: Significant differences with respect to CIT-BRG. *p* < 0.05.

**Figure 3 biology-11-00075-f003:**
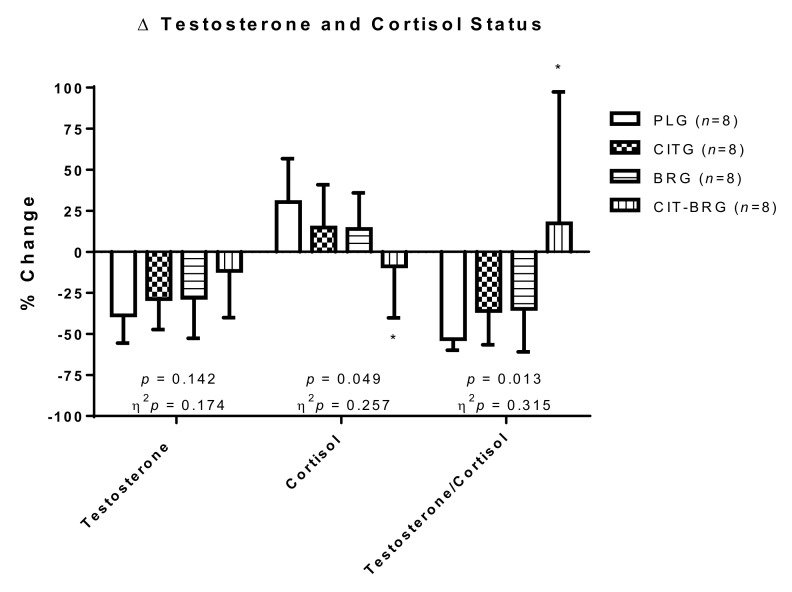
Percentage changes during the study in cortisol and testosterone hormone status and testosterone/cortisol ratio in the triathletes. Data are presented as mean ± standard deviation. Δ: ((T2 −T1)/T1) × 100. *: Significant differences with respect to PLG. *p* < 0.05.

**Table 1 biology-11-00075-t001:** Type, intensity and duration of weekly training program.

	Monday	Tuesday	Wednesday	Thursday	Friday	Saturday	Sunday
1st session	- 20 warm-up - 15 min stretching - 45 min mindfulness	- 15 min run technique skills - 45 min strength training	- 120 m ride at 50–75% VO_2max_ - 15 min cooldown - 15 min core	- 15 min run technique skills - 45 min strength training	- 75 min run at 50–75% VO_2max_ - 20 min resistance training	- 75 min swim at 75–90% VO_2max_	- 15 min run technique skills - 45 min strength training
2nd session	REST	- 30 min warm-up - 30 min run at 75–90% VO_2max_ - 15 min cooldown	REST	- 30 min warm-up - 30 min swim at 75–90% VO_2max_ - 15 min cooldown	REST	- 60 min walk	- 120 min ride at 50–75% VO_2max_ - 15 min core

**Table 2 biology-11-00075-t002:** Age and height of participants at the beginning of the study.

	PLG (*n* = 8)	CITG (*n* = 8)	BRG (*n* = 8)	CIT-BRG (*n* = 8)
Age (years)	34.01 ± 7.03	32.75 ± 7.01	32.67 ± 6.54	34.35 ± 7.95
Height (cm)	179 ± 8 cm	180 ± 9 cm	178 ± 8 cm	181 ± 6 cm

Data are presented as mean ± standard deviation.

**Table 3 biology-11-00075-t003:** Energy and macronutrient intake of triathletes during 9 weeks of study.

	PLG	CITG	BRG	CIT + BRG
Energy (kcal/kg)	45 ± 6.4	45.2 ± 6.8	44.9 ± 6.5	45.3 ± 7.2
Protein (g/kg)	1.4 ± 0.5	1.5 ± 0.7	1.4 ± 0.8	1.4 ± 0.5
Fat (g/kg)	1.4 ± 0.4	1.5 ± 0.5	1.4 ± 0.6	1.5 ± 0.6
Carbohydrates (g/kg)	7.0 ± 1.0	7.1 ± 1.2	7.1 ± 1.4	7.0 ± 1.3

Data are shown as mean ± standard deviation. PLG: placebo group, CITG: citrulline group, BRG: NO_3_^−^ group; CIT-BRG: citrulline plus NO_3_^−^ supplemented group.

**Table 4 biology-11-00075-t004:** Anthropometry and body composition outcomes of triathletes.

Group	T1	T2	*p* (T × G)	ƞ^2^*p*
**Body mass (Kg)**				
PLG	76.36 ± 7.03	76.31 ± 6.76	0.582	0.074
CITG	79.08 ± 7.36	77.70 ± 7.09		
BRG	74.11 ± 6.93	74.00 ± 6.90		
CIT + BRG	74.19 ± 11.26	74.29 ± 11.38		
**BMI (kg/m^2^)**				
PLG	24.01 ± 1.89	23.98 ± 2.03	0.407	0.115
CITG	24.52 ± 2.53	23.99 ± 2.25		
BRG	23.25 ± 1.86	23.25 ± 1.85		
CIT + BRG	22.54 ± 1.63	22.53 ± 1.59		
**Muscle mass (kg)**				
PLG	69.39 ± 5.42	69.65 ± 5.73	0.406	0.112
CITG	72.35 ± 6.21	66.05 ± 5.77		
BRG	67.38 ± 6.46	67.95 ± 6.31		
CIT + BRG	68.49 ± 9.49	68.49 ± 9.20		
**Fat mass (%)**				
PLG	9.01 ± 2.05	8.66 ± 2.14	0.121	0.203
CITG	8.45 ± 1.44	7.77 ± 1.32		
BRG	9.07 ± 2.09	8.18 ± 0.96		
CIT + BRG	7.52 ± 1.66	7.58 ± 2.17		

Data are presented as mean ± standard deviation. *p* (T × G): interaction group-by-time (*p* < 0.05) by two-factor repeated-measures ANOVA.

**Table 5 biology-11-00075-t005:** Serum EIMD markers of triathletes at T1 and T2 (after 9 weeks).

Group	T1	T2	*p* (T × G)	η^2^*p*
**Urea (mg/dL)**				
PLG	37.38 ± 6.63	38.00 ± 4.81	0.260	0.131
CITG	37.06 ± 6.92	34.58 ± 8.72		
BRG	38.38 ± 4.03	37.50 ± 2.83		
CIT + BRG	36.75 ± 8.55	41.13 ± 6.06		
**Creatinine (mg/dL)**				
PLG	0.91 ± 0.10	0.92 ± 0.10	0.601	0.063
CITG	0.93 ± 0.57	0.92 ± 0.11		
BRG	0.92 ± 0.11	0.88 ± 0.09		
CIT + BRG	0.91 ± 0.09	0.91 ± 0.11		
**AST (UI/L)**				
PLG	33.50 ± 9.06	37.13 ± 16.31	0.321	0.115
CITG	30.50 ± 9.09	24.38 ± 3.93		
BRG	39.38 ± 17.39	29.36 ± 4.79		
CIT + BRG	37.38 ± 12.02	29.75 ± 7.91		
**ALT (UI/L)**				
PLG	28.00 ± 14.25	29.63 ± 13.76	0.327	0.114
CITG	25.88 ± 9.43	22.00 ± 5.40		
BRG	36.25 ± 29.36	23.38 ± 8.45		
CIT + BRG	33.25 ± 19.83	22.63 ± 7.65		
**GGT (UI/L)**				
PLG	16.88 ± 4.55	18.50 ± 6.37	0.699	0.049
CITG	18.88 ± 8.04	19.88 ± 8.94		
BRG	15.50 ± 3.46	18.63 ± 6.41		
CIT + BRG	19.75 ± 5.92	20.25 ± 8.78		
**LDH (UI/L)**				
PLG	438.86 ± 48.13	367.38 ± 77.77	0.498	0.083
CITG	330.88 ± 90.99 ^a^	324.00 ± 71.97		
BRG	445.38 ± 247.59 ^b^	393.88 ± 35.79 *		
CIT + BRG	431.00 ± 75.05	411.88 ± 63.37		
**CK (UI/L)**				
PLG	319.50 ± 297.34	175.00 ± 51.77	0.238	0.138
CITG	327.63 ± 287.07	157.25 ± 60.78		
BRG	328.38 ± 247.59	208.88 ± 98.22		
CIT + BRG	379.38 ± 336.75	288.25 ± 209.86		

Data are presented as mean ± standard deviation. *p* (T × G): interaction group-by-time (*p* < 0.05) by two-way repeated-measures ANOVA. *: Significant differences between the two phases (T1 vs. T2) (*p* < 0.05). ^a^: Significant differences with respect to PLG (*p* < 0.05). ^b^: Significant differences with respect to CITG (*p* < 0.05).

**Table 6 biology-11-00075-t006:** Testosterone and cortisol status and testosterone/cortisol ratio of the triathletes at T1 and T2 (after 9 weeks).

Group	T1	T2	*p* (T × G)	η^2^*p*
**Testosterone (ng/mL)**				
PLG	7.66 ± 2.26	4.51 ± 1.21 *	0.116	0.188
CITG	7.77 ± 1.10	5.50 ± 1.36 *		
BRG	7.11 ± 1.26	4.92 ± 1.16 *		
CIT + BRG	7.55 ± 1.06	6.69 ± 2.50		
**Cortisol (μg/dL)**				
PLG	15.76 ± 1.34	20.37 ± 3.47 *	0.044	0.247
CITG	16.03 ± 2.48	18.20 ± 3.80		
BRG	15.89 ± 3.19	17.84 ± 3.76		
CIT + BRG	16.94 ± 2.33	15.30 ± 5.53		
**Testosterone/cortisol ratio**				
PLG	49.07 ± 15.92	22.87 ± 8.12 *	0.005	0.359
CITG	49.13 ± 7.84	31.25 ± 11.02 *		
BRG	46.95 ± 13.83	28.63 ± 8.38 *		
CIT + BRG	45.97 ± 13.16	51.66 ± 30.00 ^a^		

Data are presented as mean ± standard deviation. *p* (T × G): group-by-time interaction (*p* < 0.05) by two-way repeated-measures ANOVA. *: Significant differences between the two phases (T1 vs. T2) (*p* < 0.05). ^a^: Significant differences with respect to PLG (*p* < 0.05).

## Data Availability

No new data were created or analyzed in this study. Data sharing is not applicable to this article.

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
