# Peer review of "Long-Term Combined Effects of Citrulline and Nitrate-Rich Beetroot Extract Supplementation on Recovery Status in Trained Male Triathletes: A Randomized, Double-Blind, Placebo-Controlled Trial"

_biology, 2022, doi:10.3390/biology11010075_

Round 1

Reviewer 1 Report

Congratulations to the authors for this new version that corrects the deficiencies of the previous version, and must be accepted.

Author Response

Reviewer 1: Congratulations to the authors for this new version that corrects the deficiencies of the previous version, and must be accepted.

Authors: Thank you for your suggestions and recommendations.

Reviewer 2 Report

Dear authors,

I appreciate the effort and changes performed compared to the last submission. In this sense, the inclusion of a physical performance variable (VO2max) has increased the impact of the study. In addition, the new statistical treatment and redaction of the results section have improved the manuscript. Nevertheless, I have got the next very important concerns that haven´t be solved:

  • Sample size was not calculated. What is the statistical power of your results? I believe that they are so low.
  • Do you believe that VO2max assessed by a Cooper test is a reliable physical performance data? Did participants experienced with this test. The interpretation of this VO2max estimated must be cautelous. In fact, I propose to use the distance cover as a more suitable data than VO2max.
  • It´s not quantified training load during this intervention. Could author affirm that training load don´t affect to the results? In this sense, for continuing with this study, conclusions must be very limited and be subdued to these important limitation that could affect to the results obtained in this study.

In addition to the previous important concerns, authors must solve the next concerns: 

  • In the literature, beetroot supplementation is not abbreviated as “BEET”.
  • Nitrate and nitrite must be abbreviated along the manuscript too.
  • There is a wrong concept because EIMD is not a consequence of the decreased athletic performance. It´s necessary to be more precise.
  • Authors suggest that endocrine response post-exercise is mediated by EIMD, but it´s false. In fact, neuroendocrine response is not a consequence of EIMD. For example, an exercise that don´t cause EIMD (cycle ergometer at a severe intensity domain) influence in the neuroendocrine system, affecting to the response of both, anabolic and catabolic hormones.
  • Authors say that NO would allow greater protein synthesis [14]. For making this affirmation is necessary to include a study that have assessed this relation directly.
  • Do authors believe that the effect of NO in the testis of animals could be reproductible in humans mediated by the response to the supplements studied? This mechanism continue as very speculative and must be revised once again.
  • There is not any justification in the introduction for mentioning a possible combined effect of creatine and HMB, specially because the ergogenic effect of HMB could be influenced with studies that present conflict of interests. In any way, it´s not important in this study and it´s not associated.
  • In all the interpretation of the results, authors haven´t considered a possible difference in the training load between groups.

Author Response

Point-by-Point Response to Reviewer’s Comments

We would like to sincerely thank to the reviewers for their helpful recommendations. We have seriously considered all the comments and carefully revised the manuscript accordingly. Revisions are highlighted in navy through the manuscript in order to indicate where changes have taken place. We feel that the quality of the manuscript has been significantly improved with these modifications and improvements based on the reviewers’ suggestions and comments. We hope our revision will lead to an acceptance of our manuscript for publication in Biology.

In advance,

Kind regards

Reviewer 2

Dear authors,

I appreciate the effort and changes performed compared to the last submission. In this sense, the inclusion of a physical performance variable (VO2max) has increased the impact of the study. In addition, the new statistical treatment and redaction of the results section have improved the manuscript. Nevertheless, I have got the next very important concerns that haven´t be solved:

REVIEWER: Sample size was not calculated. What is the statistical power of your results? I believe that they are so low.

AUTHORS: Thank you for your interest. The actual statistical power using G*power 3.1.9.4 software is 0.5870359 based on the following data:

[17] -- Saturday, December 18, 2021 -- 09:56:40

F tests - ANOVA: Repeated measures, within-among interaction

Analysis:      Post hoc: Compute achieved power

Input:            Effect size f                                      =    0.25

                        α err prob                                        =    0.05

                        Total sample size                           =    32

                        Number of groups                         =    4

                        Number of measurements          =    2

                        Corr among rep measures           =    0.5

                        Nonsphericity correction ε          =    1

Output:         Noncentrality parameter λ         =    8.0000000

                        Critical F                                          =    2.9466853

                        Numerator df                                 =    3.0000000

                        Denominator df                             =    28.0000000

                        Power (1-β err prob)                     =    0.5870359

In this line, we have assumed that this is a limitation and have added a sentence in the limitation section: “On the other hand, sampling using a convenient, non-probabilistic sampling procedure may produce results that are not representative of the rest of the population. These limitations may underrepresent the results and may affect study outcomes. For this reason, the results should be taken into account in the context of the study.”

REVIEWER: Do you believe that VO2max assessed by a Cooper test is a reliable physical performance data? Did participants experienced with this test. The interpretation of this VO2max estimated must be cautelous. In fact, I propose to use the distance cover as a more suitable data than VO2max.

AUTHORS: Thank you very much for your recommendation. We have substituted VO 2 max variable for distance covered. We have also included the sentence that the athletes were familiar with this test, given that they usually use this test throughout the season  

REVIEWER: It´s not quantified training load during this intervention. Could author affirm that training load don´t affect to the results? In this sense, for continuing with this study, conclusions must be very limited and be subdued to these important limitation that could affect to the results obtained in this study.

AUTHORS: All athletes rigorously performed the same training methodology and thus, all of them were exposed to the same training load in terms of type, intensity, and duration of exercise: 15 h/week, 6 days/week during the 9 weeks.

Table 1. Type, intensity, and duration of weekly training program.

Monday

Tuesday

Wednesday

Thursday

Friday

Saturday

Sunday

1 st session

- 20 warn up

- 15 min stretching

- 45 min mindfulness

- 15 min run technique skills

- 45 min Strength training

- 120 m Ride  at 50-75 % VO2max

- 15 min could down

- 15 min core

- 15 min run technique skills

- 45 min Strength training

- 75 min Run at 50-75 % VO2ma

- 20 min resistance training up

- 75 min  Swim at 75-90 % VO2ma

- 15 min run technique skills

- 45 min Strength training

2º session

REST

- 30 min warn up

- 30 min Run at 75-90 % VO2ma

- 15 min could down

REST

- 30 min warn up

- 30 min Swim at 75-90 % VO2ma

- 15 min could down

REST

- 60 min walk

- 120 min Ride at 50-75 % VO2ma

- 15 min core

In addition to the previous important concerns, authors must solve the next concerns: 

REVIEWER: In the literature, beetroot supplementation is not abbreviated as “BEET”.

AUTHORS: Thank you for your recommendation. The authors have changed BEET by BR.

REVIEWER: Nitrate and nitrite must be abbreviated along the manuscript too.

AUTHORS: Thank you for your recommendation. Nitrate and nitrite have been abbreviated as NO3- and NO₂

REVIEWER: There is a wrong concept because EIMD is not a consequence of the decreased athletic performance. It´s necessary to be more precise.

AUTHORS: Thank you for your comment. We have been reviewing the document and have not found the sentence that indicates your comment. However, if the reviewer tells us exactly where we say this statement, we will be happy to change it.

On the other hand, although it is true that the detriment of sports performance is a consequence of many factors, there is sufficient evidence that the greater EIMD is relationship with lower athletic performance.

Coso, J. D., Gonzalez-Millan, C., Salinero, J. J., Abian-Vicen, J., Soriano, L., Garde, S., & Perez-Gonzalez, B. (2012). Muscle damage and its relationship with muscle fatigue during a half-iron triathlon.

Del Coso J, González C, Abian-Vicen J, Salinero Martín JJ, Soriano L, Areces F, Ruiz D, Gallo C, Lara B, Calleja-González J. J Sports Sci. 2014;32(18):1680-7. doi: 10.1080/02640414.2014.915425. Epub 2014 May 13. PMID: 24825571

Burt, D. G., & Twist, C. (2011). The effects of exercise-induced muscle damage on cycling time-trial performance. The Journal of Strength & Conditioning Research, 25(8), 2185-2192.

Twist, C., & Eston, R. (2005). The effects of exercise-induced muscle damage on maximal intensity intermittent exercise performance. European journal of applied physiology, 94(5), 652-658.

Highton, J. M., Twist, C., & Eston, R. G. (2009). The effects of exercise-induced muscle damage on agility and sprint running performance. Journal of Exercise Science & Fitness, 7(1), 24-30.

Byrne, C., Twist, C., & Eston, R. (2004). Neuromuscular function after exercise-induced muscle damage. Sports medicine, 34(1), 49-69.

…..

REVIEWER: Authors suggest that endocrine response post-exercise is mediated by EIMD, but it´s false. In fact, neuroendocrine response is not a consequence of EIMD. For example, an exercise that don´t cause EIMD (cycle ergometer at a severe intensity domain) influence in the neuroendocrine system, affecting to the response of both, anabolic and catabolic hormones.

AUTHORS: Thank you for your comment. We agree with your affirmation, however, we have been reviewing the document and have not found the sentence that indicates what you comment. However, if the reviewer tells us exactly where we say this statement, we will be happy to change it.

REVIEWER: Authors say that NO would allow greater protein synthesis [14]. For making this affirmation is necessary to include a study that have assessed this relation directly.

AUTHORS: Thank you for your comment. Based on the 14 reference we have changed “greater protein synthesis” by “better muscle fueling”.

REVIEWER: Do authors believe that the effect of NO in the testis of animals could be reproductible in humans mediated by the response to the supplements studied? This mechanism continue as very speculative and must be revised once again.

AUTHORS: Thank you for your comment. Considering that the authors have not directly measured anything in muscle, all possible effects discussed are speculative, including those that have occurred in animal models and have not been tested in humans. In this regard, we have included the following to make it clear that this hypothesis is speculative: “Consequently, although this mechanism is speculative, increasing NO could improve blood flow in the testis and promote testosterone synthesis by vasodilator effect [14,19]…”.

Consequently, although this hypothesis is speculative, increasing NO could be successful in maintaining an anabolic state, decreasing metabolic stress [2,9].

In addition, we include in the limitations section the following text: “the effects that both supplements used could have on the muscle were speculative because no evaluation has been included in this regard.”

REVIEWER: There is not any justification in the introduction for mentioning a possible combined effect of creatine and HMB, specially because the ergogenic effect of HMB could be influenced with studies that present conflict of interests. In any way, it´s not important in this study and it´s not associated.

AUTHORS: Thank you for your recommendation. The authors have deleted the terms HMB and creatine. However, the sentence tries to justify that other studies (our group studies with our main research line and there is not conflict of interest), have shown that co-supplementation in the long term produced synergistic effects with respect to their individual supplementation. That is one of the reasons for testing this co-supplementation (BR and CIT).

REVIEWER: In all the interpretation of the results, authors haven´t considered a possible difference in the training load between groups.

AUTHORS: The training loads were the same for all participants: “all of them were exposed to the same training load in terms of type, intensity, and duration of exercise: 15 h/day, 6 days/week during the 9 weeks.”

Table 1. Type, intensity, and duration of weekly training program.

Monday

Tuesday

Wednesday

Thursday

Friday

Saturday

Sunday

1 st session

- 20 warn up

- 15 min stretching

- 45 min mindfulness

- 15 min run technique skills

- 45 min Strength training

- 120 m Ride  at 50-75 % VO2max

- 15 min could down

- 15 min core

- 15 min run technique skills

- 45 min Strength training

- 75 min Run at 50-75 % VO2ma

- 20 min resistance training up

- 75 min  Swim at 75-90 % VO2ma

- 15 min run technique skills

- 45 min Strength training

2º session

REST

- 30 min warn up

- 30 min Run at 75-90 % VO2ma

- 15 min could down

REST

- 30 min warn up

- 30 min Swim at 75-90 % VO2ma

- 15 min could down

REST

- 60 min walk

- 120 min Ride at 50-75 % VO2ma

- 15 min core

Reviewer 3 Report

I’ve read with attention the paper of Burgos et al. that is potentially of interest. The background and aim of the study have been clearly defined, even if the introduction is very long and somewhat unfocused. The authors should limit their introduction to the main topic of the experimental part of the paper. The methodology applied is overall correct, the results are reliable and adequately discussed. However, the discussion section is also very long (in particular considering the very small patient sample on which the study has been carried out) and somewhat unfocused. It could be improved.  

Author Response

Reviewer 3: I’ve read with attention the paper of Burgos et al. that is potentially of interest. The background and aim of the study have been clearly defined, even if the introduction is very long and somewhat unfocused. The authors should limit their introduction to the main topic of the experimental part of the paper. The methodology applied is overall correct, the results are reliable and adequately discussed. However, the discussion section is also very long (in particular considering the very small patient sample on which the study has been carried out) and somewhat unfocused. It could be improved. 

AUTHORS: Thank you very much for your recommendation. Although we agree with the reviewer's comment, we have not dared to delete anything from it because it has included the reviewers' suggestions and comments for 3 rounds. However, if the reviewer feels that a specific section or sentence should be deleted, we would be happy to do so.

Round 2

Reviewer 2 Report

Dear authors,

Thank you for your answer, specially related to the concerns about meethodology and limitations of the study. Now, the manuscript present sufficient quality for being published.

This manuscript is a resubmission of an earlier submission. The following is a list of the peer review reports and author responses from that submission.

Round 1

Reviewer 1 Report

I want to start by congratulating the authors for this manuscript, which is original although it still has a lot of room for improvement.
In material and methods, the medical history questionnaires used should be referenced or added as supplementary material, although in this section they indicate that they do not have any disease, but they are not defined or referenced.
In this same section of participants, the reference CLR-0020 is indicated, and it seems that it may be a code of some professional association of dieticians / nutritionists, but I have not found any other reference. This is important, because they indicate that personalized diets are made, but neither macro nor micronutrient parameters used are indicated, nor is there any reference that indicates it. On the other hand, the protocols used in the anthropometric study or the FFQs used are cited and referenced with bibliography.
A book with more than 500 photographs is also indicated, which I imagine will be the book "Laminas de alimentos de porciones a tamaño real" by Iva Marques-Lopes and Giuseppe Russolillo, which should also be cited.
In table 1 of results, CITG and CIT + BEETG appear in the table and in the legend I think it appears as CMG and CM-BEETG.
In table 2, in the body mass (Kg) section, the BEETG and CIT-BEETG groups indicate the authors that there are significant differences between T1 and T2, but the data are practically the same, with 74.11 ± 6.93 vs 74.00 ± 6.90 for BEETG , and 74.19 ± 11.26 and 74.29 ± 11.38 for CIT + BEETG respectively. The results are very similar to have significant differences with such a small n. The same happens for these groups in BMI, so you have to review in depth all the tables and results.
It would also be desirable for the authors to indicate in results only the values ​​with significant differences, and the rest of the results can be indicated as supplementary material. This makes it easier to follow the manuscript.
Regarding the discussion, there is no table or figure of results referenced in the discussion, or mean values ​​of two or more groups. For this reason, the authors should elaborate a discussion according to their results, and where they indicate their most significant values ​​according to other similar studies, in addition to giving value to their study, and that guide the readers to the final conclusions.
Regarding the limitations, at present it is necessary to carry out studies in men and women to improve the impact of the publications, even if it is with a low n.
For all this, the article has to be rewritten and rejected in this version.

Reviewer 2 Report

Dear authors,

This study present as objective to analyse the effect of beetroot, citrulline and co-ingestion of both supplements on exercise-induced muscle damage (EIMD) markers and anabolic/catabolic hormones in trained triathletes. The hypothesis of authors is that the co-ingestion of beetroot and citrulline improve endogenous recovery and exercise adaptations more than isolated supplementation and placebo. However, according to the experimental design, it´s impossible the next:

  • To analyse the effect of supplementation on recovery. How did authors assess exercise recovery? There is not any assessment of exercise recovery. For that, authors must standardize the training load (at least the lasts days of training) and include some functional tests.
  • To analyse the effect of supplementation on training adaptions. How did authors assess training adaptations? In the variable analysed there is not any indicators of training adaptation.

Based on the two previous comments, the objectives of the study cannot be answered. Therefore, the experimental design of this study isn´t suitable and it´s bad conceptualized. In addition, independently of the unsuitable objectives, it exists a factor that could affect to the results founds. In this way, authors haven´t quantified the training load. Previous studies have informed of ergogenic effect of beetroot and citrulline on endurance. Considering that authors haven´t include any physical test in the pre- and post-intervention, authors must consider the different experimental condition could affect to the training load, increasing it in some groups. If the training load was higher in some groups, the interpretation of the biochemical variables is different. This interaction makes that the results reported are inconsistent.

The unsuitable objectives as the variables data and the interaction didn´t consider about the training load as confounder makes that this study cannot be accepted. However, there is a lot of bad concepts in the text that could be implied the incorrect design proposed by authors. In this way, I mark the next major concerns:  

  • In the literature, beetroot supplementation is not abbreviated as “BEET”. The same is for nitrate which is abbreviated as “NO3-”.
  • In the first sentence, it exists a wrong concept because authors indicated that strenuous exercise followed by an inadequate recovery causes muscle damage. This is false, because an important mechanical load (for example, an acute training session based on eccentric contractions) provokes muscle damage.
  • At the end of the first sentence, authors associate muscle damage post-exercise with a decreasing in athletic performance. However, in the post-exercise phase the diminution of force production is considered fatigue. There are two wrong concepts in the first sentence of the manuscript.
  • In the second sentence, authors suggest that endocrine response post-exercise is mediated by EIMD, but it´s false. In fact, neuroendocrine response is not a consequence of EIMD. For example, an exercise that don´t cause EIMD (cycle ergometer at a severe intensity domain) influence in the neuroendocrine system, affecting to the response of both, anabolic and catabolic hormones.
  • Why authors include information related to “density”. Is it important for your study the training density?
  • It´s not necessary to explain the satellite cells. Readers know it.
  • Authors say that NO would allow greater protein synthesis [14]. Nevertheless, reference 14 don’t support this affirmation.
  • Do authors believe that the effect of NO in the testis of animals could be reproductible in humans mediated by the response to the supplements studied? This mechanism is very speculative and cannot be supported.
  • In the explanation of the citrulline effects are not related the mechanism of action with the ergogenic effects indicated.
  • Reference 30 doesn´t justify that must be studied the co-ingestion of citrulline and beetroot supplementation.
  • There is not any justification in the introduction for mentioning a possible combined effect of creatine and HMB, specially because the ergogenic effect of HMB could be influenced with studies that present conflict of interests. In any way, it´s not important in this study and it´s not associated.
  • In the section “2.2. Experimental Protocol and Evaluation Plan”, information related to the characterization of the different experimental group could be more readable in a table.
  • Did authors calculate the sample size necessary for this type of study?
  • Could authors be ensured that participants cannot differentiate between the different supplements and the placebo used?
  • Based on the effect of oral bacteria on the reduction of nitrite from nitrate, did authors include any caution (for example, limit tooth brushing).
  • What was the timing of the ingestion of the different supplements the day of the experimental sessions and during the intervention? This aspect is very important because could interfere in the results found.
  • Why authors include a t-student test for comparing pre vs post? The most suitable statistical treatment is to include a two-way AVOVA-RM with the factor time.
  • How did authors control training load during the intervention. In this sense, a possible effect of the different experimental condition could increase training load which imply in a different interpretation of the results reported.
  • In the results section, authors mustn´t repeat results reported in the tables under the text (for example, all the medium and standard deviations of the variables included  in the table 3).
  • It´s not necessary to include the meaning of the effect size because it´s explained in the statistical treatment section.
  • Regarding to the interpretation to “triathletes showed a significantly better endogenous adaptations in CIT-BEETG by an prevent a decline in cortisol and better T/C ratio” is not correct. Based on these two parameters without any physical variable and/or physical performance, you cannot say it.
  • In all the interpretation of the results, authors haven´t considered a possible difference in the training load between groups.
  • All the explanation about cortisol and testosterone without the assessment of physical performance is not correct because authors cannot check if the results in the biochemical parameters are concomitant to enhancement or impairment on the physical performance. In fact, authors indicate “The balance between training loads and recovery are key factors in increasing athletic performance [4]”. If you didn´t consider training load, the possible parameters of recovery are deficiencies of any meaning.
  • Regarding to all the possible effect of the two supplements used in the muscle, all of the are speculative because authors haven´t included any assessment about it.
  • Authors explain a possible effect of beetroot on anabolic/catabolic status that it´s not included in the cite used.
  • Conclusions of the study are incorrect because authors cannot make any conclusion about exercise recovery for the comments performed during this revision.

Reviewer 3 Report

I’ve read with attention the paper of Burgos et al. that is potentially of interest. The authors assessed the effect and degree of potentiation of the
long-term (9 weeks) mixture of 3 g/day of citrulline plus 2.1 g/day of beetroot (300 mg/day of NIT) supplementation on exercise-induced muscle
damage markers (urea, creatinine, AST, ALT, GGT, LDH and CK) and on anabolic/catabolic hormone (testosterone, cortisol and testosterone / cortisol ratio (T/C)) in male trained triathletes.  The background and aim of the study have been clearly defined. The methodology applied is overall correct, the results are reliable and adequately discussed. I've only a main concern. If it is true that the it is difficult to enrol large subject samples in this context, however the study risk to be strongly underpowered, while no power estimation nor sample size calculation has been reported. Moreover, it is not clear why it was needed to study a long-term exposition in spite to carry out a cross-over trial, that could have a much more stronger power. This has to be more deeply discussed. 

Reviewer 4 Report

Nitric oxide (NO) is a gaseous signalling molecule involved in a variety of physiological functions throughout the body. The first pathway for NO production is endogenous via the citrulline-arginine-NO pathway requiring the activity of the nitric oxide synthase (NOS) enzymes. The alternative pathway uses nitrate and nitrite brought by water and food to produce NO. It has been shown that oral citrulline administration as well as dietary nitrate supplementation can modify nitrate and nitrite concentration, potentially increasing NO bioavailability (Bescos et al, sports med 2012). As a consequence, several studies have investigated the potential benefits deriving from these supplementation strategy on different physiological functions NO-related. This study aimed to effect of the long-term (9 weeks) combination of citrulline and beetroot extract on exercise-induced muscle damage markers and on anabolic/catabolic hormonal profile in male trained triathletes. Thirty-two subjects were divided into four groups and supplemented by citrulline, nitrate-rich beetroot extract, citrulline + beetroot extract or placebo group for 9 weeks. Before and after the intervention, exercise-induced muscle damage markers (urea, creatinine, AST, ALT, GGT, CK and LDH) and catabolic/anabolic hormonal profile (testosterone/cortisol) were measured in the blood. According to authors conclusions, EIMD markers were not affected by the intervention whereas the combination of citrulline and beetroot extract prevented a decline in cortisol and improved testosterone/cortisol ratio. Thus, the authors stated that the combined use of citrulline and beetroot extract can promote faster muscle recovery from high-intensity activity, without preventing EIMD.

Although the topic investigated by the authors is interesting, I think this manuscript has strong limitations:

- the hypothesis is not justified by data presented from the literature. In the introduction, the authors state that increasing NO bioavailability can affect exercise-induced muscle damage. However, the authors lack to take into consideration solid studies present in literature. For example, in their recent systematic review and meta-analysis, Jones L and collegues (DOI: 10.1080/19390211.2021.1939472) examined whether dietary nitrate supplementation attenuates exercise-induced muscle damage (EIMD) and concluded that there is little evidence that nitrate can modify markers of oxidative stress or inflammation. Additionally, the authors aim to investigate the effects of citrulline + nitrate-rich extract on testosterone/cortisol concentrations because increasing NO may improve blood flow in the testis and promote testosterone synthesis by vasodilator effect. However, Leydig cells production of testosterone is limited only for severe reduction of blood flow whereas in normal conditions oxygen supply is not limiting.

- the methodological approach has important pitfalls. The authors administered citrulline and/or inorganic nitrate to increase NO bioavailability but there is no evidence about the effects of these interventions on NO pathways. The authors should have taken into consideration measurements such as plasma nitrate and nitrite concentration. Additionally, in the present work EIMD was characterized by non-specific blood markers (urea, creatinine, AST, ALT, GGT, LDH and CK), and it is not clear why the authors decided not to include other important biomarkers (e.g. myoglobin) or functional measurements (e.g. strength, muscle pain).

- statistics should be revised. The paragraph about statistics is not clear. Why did the authors performed Student's t-tests for differences from T1 to T2 in each group whereas comparisons among groups were assessed by Bonferroni post-hoc after a two-way repeated measures analysis of variance? Are the authors sure that the additive method approach was the right choice? From the best of my knowledge, this approach is usually utilized for non-parametric parameters. The power calculation is overly vague and it would be useful to have included more information in terms of the standardized mean difference and what the authors consider is a minimally clinically significant difference (i.e. critical difference) of their end-outcome variables.

- values of testosterone. According to recent guidelines from the American Urological Association (AUA), a testosterone level of at least 300 ng/dL is normal for a man. The normal range for morning T in a male is between 300 and 1,000 ng/dl and likewise, hypogonadism has been defined as total T < 300 ng/dL by the Endocrine Society clinical practice guidelines (Harman SM, Metter EJ, Tobin JD, Pearson J, Blackman MR. Longitudinal effects of aging on serum total and free testosteronal levels in healthy men. Baltimore longitudinal study of aging. J Clin Endocrinol Metab. (2001) 86:724–31. doi: 10.1210/jcem.86.2.7219). In this study, testosterone values ranged between 4 and 7 ng/dl. How is it possible? Are the authors sure about these values?

- the conclusion are not supported by the results. “Since 3 g/day of CIT plus 2.1 g/day of BEET (300 mg/day of NIT) for 9 weeks could advance muscle and endogenous recovery, supplementation phases could be considered in the intensive training phases”. In my opinion, it is not possible to consider a change in cortisol level, without any difference in other markers, among the interventions as a marker of improved recovery. Recovery from exercise or physiological adaptations to training are very complex processes and cannot be characterized by a unique blood marker. I advise the authors to revise their work and pay attention in drawing such strong conclusions or giving practical suggestions.

- There are numerous grammatical errors though the text and a native English revision should be performed.